# Activity Demands and Speed Profile of Young Female Basketball Players Using Ultra-Wide Band Technology

**DOI:** 10.3390/ijerph17051477

**Published:** 2020-02-25

**Authors:** María Reina, Javier García-Rubio, Sergio J. Ibáñez

**Affiliations:** 1Training Optimization and Sports Performance Research Group (GOERD), Sport Science Faculty, University of Extremadura, 10005 Caceres, Spain; sibanez@unex.es; 2Facultad de Educación, Universidad Autónoma de Chile, Santiago 7500912, Chile

**Keywords:** basketball, gender, adolescent, profile, performance

## Abstract

Performance profiles have begun to be identified as extremely useful in order to help coaches individualize training according to the age and gender of athletes. Therefore, the aim of this study was to determine the activity demands and speed profile of U18 female basketball players during competitive matches. Time variables (real and playing time), distance variables (distance performed, distance in speed zones, high intensity distance and distance covered sprinting) and speed variables (number of sprints, sprint duration, maximum speed and average speed) were recorded from forty-eight players belonging to four teams (13 guards, 22 forwards and 13 centers). WIMUPRO^TM^ inertial measurement units with ultra-wide band (UWB) indoor-tracking technology recorded six matches during final four in the season 2018/2019. A one factor ANOVA with Cohen’s effect sizes (*d*) were used to identify the differences between groups (playing position and match day). Distance per minute (123.96 vs 112.67 m), high intensity distance per minute (15.48 vs 14 m), running distance (403.2 vs 541.28 m) and average speed (5.05 vs 5.41 km/h) were significantly higher on day 3 than 1, respectively. About playing position, forwards played more minutes during games, so covered a greater distance, more sprints and high intensity actions than the rest. In spite of fatigue, day 3 showed a greater intensity than day 1, therefore, the last day was the crucial one for the teams in the tournament. Forwards when playing more minutes obtain higher absolute values ​​but not per minute which could mean a lower performance of the team.

## 1. Introduction

The analysis of demands during competition in team sports is becoming increasingly popular. The existing research in basketball focuses mainly on professional adult players, revealing a lack of knowledge on performance in the formative stages. The evaluation of adolescent players is of vital importance, especially in U18 players, in order to support a proper transition to the adult stage [1,2]. This period is characterized as a time when players can tolerate high training loads and demands in competition, as well as improve their levels of technical and tactical performance [3]. In this regard, it would be important to add knowledge on the monitoring of physical performance in youth basketball players nearing transition to a senior level and compete at high level in professional leagues. Research in sports performance has increased at an accelerated pace, partly due to new technologies that open access to new data and data analysis. In this field, the analysis of performance indicators, specifically profiling to describe and understand sports behaviours, is important [4]. Previous research has evaluated profiles on physiological and anthropometric characteristics in elite basketball players, although descriptive data on the characteristics of basketball players are lacking [5].

According to Buchheit and Simpson [6] monitoring players using tracking devices aims at assessing external load demands during competition, optimizing load patterns, improving performance and preventing injury. With respect to the optimization and improvement of performance using load patterns, performance profiles have begun to be identified with the purpose of helping coaches to individualize work during training according to the type of player, thus improving players’ physical fitness [7]. In indoor sports new methods of measurement like inertial movement units (IMUs) and ultra-wide band radio frequency (UWB) has begun used. With this type of tools, it is possible to generate objective and reliable information both on body load from the accelerometers and locomotion using tracking devices.

In previous investigations, match activity in basketball has been classified into the following different categories: Standing/walking (0–1.67 m·s^−1^); Jogging (1.1–3.3 m·s^−1^); Running (3–7 m·s^−1^) and Sprinting (>7 m·s^−1^) [8,9,10,11,12,13]. According to the review by Stojanović [14] the percentages of metres performed in each speed zone are established according to the following format: Standing/walking: 23.4–66.3%; Jogging: 5.6–36.3%; Running: 4.5–33.2%; and Sprinting: 0.3–8.5%. These range of values differ mainly due to the lack of specificity and individualization in the different studies. In women’s basketball, it has been reported that the players cover an average of 5214 ± 315 m per game. The intensity at which they work during the match is 30.2–39.3% walking, 12.8–24% jogging, 4.9–11% running, and 0.6–7.8% sprinting [8,10,11,13] being different from the results shown above in general basketball. Stojanović [14] state that both the level and the playing position influence the activity demands and physiological responses experienced by basketball players during match play. For example, in the case of the guards and elite players they record higher load values than forwards, centres and players of a lesser level. Therefore, it is important for greater specificity in data collection that samples are individualised according to different factors like gender, category, playing position, level of play, match day, play period, play time, etc. 

This information presents important conclusions for sports training. Therefore, the aim of this study was to determine the activity demands and speed profile of U18 female basketball players during competitive matches. This study categorised speed by match day and playing position using tracking devices and UWB technology. 

## 2. Materials and Methods

### 2.1. Participants

All the players and coaches were informed about the research protocol, requisites, benefits, and risks, and the legal guardian of each player written consent was obtained before the start of the study conforming to the Code of Ethics of the World Medical Association (Declaration of Helsinki) which was approved by the Ethics Committee of the University of Extremadura (nº 1234/2019). Forty-eight players belonging to four teams that classified for a final four junior women’s championship (Guards: *n* = 13, 168.62 ± 5.94 cm; Forwards: *n* = 22, 176.87 ± 6.04 cm; Centres: *n* = 13, 183.77 ± 4.71 cm). WIMUPRO^TM^ inertial measurement units with ultra-wide band (UWB) indoor-tracking technology recorded six matches during final four in the season 2018/2019. Of which, the top three classifieds will qualify to play the national championship. 

### 2.2. Variables

The matches were analysed according to playing positions (Guard, Forward and Centre) and match day (Day 1, Day 2, Day 3) in order to provide more specific information for coaches and physical trainers. The variables analysed were: Time variables (real time and playing time); Distance variables (total distance, explosive distance, percentage and metres performed in each intensity zone, and distance performed in sprints); Speed variables (number of sprints; sprint duration; maximum and average speed).

Time variables:Real time (RT): Time in minutes that each match lasted. Rest periods between quarters and timeouts were excluded from the study.Playing time (PT): Time collected in the stats when the ball was alive for each player.

Distance variables:Distance (D): Number of metres covered by the players while on the field.Speed Zones (SZ): The total distance covered was classified into the following speed zones of five intensity levels (Table 1): Standing, Walking, Jogging, Running and Sprinting, analysing the metres covered in each group and the percentage based on the total metres.Explosive distance (ED): Distance covered at over >10.2 km/h.Distance covered sprinting (HID): High intensity distance covered at over 14.4 km/h.

Speed Variables:Number of sprints: Number of times the player exceeded a speed of 14.4 km/h.Sprint duration: Time in seconds spent in each sprint repetition.Maximum speed: Maximum speed reached by a player during the match (km/h).Average speed: Average speed of a player’s locomotion during the match (km/h).

### 2.3. Match Analysis

Activity demands and speed data were obtained from a positional tracking system. Each of the six competitive matches was recorded using inertial devices (IMUs) and ultra-wide band (UWB) systems placed on the court. A protocol to analyse the match was followed and repeated at the beginning of each day. The UWB system was calibrated one hour before the start of the games and the WIMU^TM^ inertial devices were synchronised to the UWB system through the ANT + technology. Each player was fitted with the inertial device 20 minutes before the start of the match so that there was a period of familiarization during the warm-up. Once the match started, total and live times were calculated using the SVIVO^TM^ software; with total time referring to all of the time that a player was on court, including all stoppages in play, but excluding breaks between quarters and timeouts. Live time corresponded to the time when the clock was running and the player was on court and also short moments in which the player was active during out-of-bounds [14,15]. Only the players on the court were analysed. The variables were measured in two different ways: Accumulatively (the sum of all actions performed in the total game time) and according to intensity (actions per minute of play). A descriptive and inferential analysis was made according to match day and specific position.

### 2.4. Equipment

Each player was equipped with a WIMUPRO^TM^ inertial device that was turned on and placed in a specific custom vest fitted tightly to the body, as is typically used in games, located on the posterior side of the upper torso (Figure 1). The UWB system was adjusted to the reference field before the start of the investigation, by going around the perimeter of the field for it to be recognised as the reference system [15]. This reference system was established with the actual measurements of the field (28 × 15 m) and was composed of 6 antennae placed in a hexagon around the playing field (Figure 2), enabling the automatic synchronisation of time and positioning data in the software (SPRO^TM^). The SVIVO^TM^ software automatically analyses all the data gathered by the inertial device and sends it to the computer screen in real time. 

### 2.5. Statistical Analysis

The distribution of the data was analysed with the Kolmogorov-Smirnov test [16], to select the subsequent statistical procedure. A descriptive analysis of the data was performed of all the collected variables in the study on the competition. A K-means cluster analysis was used to distribute intensity ranges of speed. Next, a one factor ANOVA, with the effect size according to Cohen’s d, was used to identify the differences between groups (playing position and match day) by the effect magnitude of intensity. Effects sizes were calculated by Cohen’s d from the F-test where effect sizes of 0.20 were considered as small, 0.50 as medium and 0.80 as large [17]. Statistical analyses were performed using SPSS v.21 software (IBM Inc., Chicago, IL, USA). Statistical significance was set at *p* < 0.05.

## 3. Results

### Descriptive results

Firstly, the evolution of the analysed variables was studied during the competition, in this case three days (Table 2). In congested-fixture tournaments it has been observed that the matches last longer as the tournament progresses leading to a longer distance covered by the teams in the last match. Regarding locomotion intensity, during the first days of competition a greater percentage of metres was covered at low intensity, while on the last day of competition (the most crucial) there was a greater percentage at higher intensity. On the last day there were more sprints of longer duration, performed at a slightly higher average velocity than the other days.

Statistically significant differences were found according to the match day and the distance covered per minute and average speed (*p* = 0.000; *p* = 0.000 and *p* = 0.009) between Day 1 and the rest, with the results obtained on Day 1 being significantly lower. The distance covered at running intensity was also significantly higher on Day 3 than on Day 1. Secondly, competitive demands were analysed according to the playing position of each player during the competition (Table 3). The forwards were the ones that played the longest time and thus recorded the greater distance covered. Regarding intensity, all the positions recorded a greater percentage of metres walking. However, the forwards covered a greater percentage of metres in the higher intensity zones (Jogging and Running). The forwards performed more sprints that the rest of the positions, but the guards performed more sprints per minute played, indicating a greater demand.

Statistically significant differences were found with respect to playing position. The forwards played more minutes during the matches (Real Time, *p* = 0.0001; Playing Time, *p* = 0.004) so that they also covered a higher number of metres compared to the guards and centres. There were therefore statistically significant differences in the variables of distance covered, explosive distance, metres covered in intense zones and metres covered in sprints (*p* < 0.001). Furthermore, statistically significant differences were found regarding the number of sprints performed, and were higher in the forwards compared to the guards. However, the guards performed more sprints per minute.

## 4. Discussion

The objective of this study was to determine the speed profile of U18 women’s basketball players during competitive matches. This study categorised locomotion demands, speed and distance, by match day and playing positions, finding significant differences. The main results revealed that on Day 1 there were lower values of distance covered and mean velocity compared with the rest of the days. All the playing positions covered a greater percentage of metres walking; however, it was the forwards who covered a higher percentage of metres in the high intensity zones, also performing more sprints. 

Ziv and Lidor [18] carried out a review of the literature in which they collected data on the demands of activity in women’s basketball through time motion analysis. Later on, more objective techniques began to be used to analyse competitive demands due to the errors inherent in time motion analysis. To solve this problem external load is currently measured using UWB technology which is considered a viable tool in team sports. Several authors have used this technology to determine external load in basketball players during sports competition [15].

Regarding the match day, Ibáñez et al. [19] mention the necessity to study the current characteristics of basketball tournaments with consecutive matches. Higher values have been found for playing time, distance covered, % running, number of sprints, duration and metres covered at high intensity in the third and last match of the competition. Pino-Ortega and Rojas-Valverde [20] also found a greater demand in the third match of a men’s U18 tournament, recording higher values in distance covered although significant differences were not found. They justified this mainly due to the use of the unlimited substitution rule. 

With respect to the different playing positions, it has been shown that guards perform more high intensity activities than forward and centres [21]. In women’s basketball, the guards spend more time in the standing/walking zones and the forwards are the ones that spend more time sprinting [8]; and comparing with the present study, it can be seen that the forwards spend less time standing/walking (44.35%) and more time sprinting (11.49%) recording very similar values. The centres are normally the ones who have a lower fitness level compared to the rest of the positions. 

Regarding the available literature on distance accumulated for female basketball players during matches, it was found that the distance covered in most studies was between 4 and 6 km during 40 min matches, mainly during the playing time [8,10,22]. Oba and Okuda [22] stated that in women’s basketball the forwards are the ones who cover greater distances compared to the guards and centres, in contrast to men’s basketball where the guards covered significantly more distance than the rest [23].

The distribution of the different ranges of running speed has been studied mainly in men’s elite basketball. Therefore, these references are also taken for women’s basketball [8,11,24]. For this reason, in this study the speed zones were individualised and similar ranges were found for the lower intensity zones, but the sprinting differed as it was classified as running at more than 4 m·s^−1^. In the literature, and more precisely in studies on men’s basketball, the sprinting zone is defined as speeds of more than 7 m·s^−1^ [8,9,10,11,12,13].

The present study recorded 45.15 ± 2.28% Standing/Walking; 22.37 ± 0.74 Jogging; 21.71 ± 1.35 Running; and 10.77 ± 0.23 Sprinting. All the values are within the range established in the bibliography, but make it possible to individualise into one single category. To correctly analyse the competitive demands, it is important to use a clear method of data processing, mainly in the play time analysis. Scanlan et al. [8] showed different results with regard to distance covered by the women players, recording 5215 ± 314 m in live playing time and 7039 ± 446 m in total playing time. This is why in this study the total match time has been analysed (real time) as well as the time that each player was on the court during playing time. Furthermore, each variable is weighted per minute of play. 

Several factors can make the competitive demands different as a function of the match day, especially in tournaments, under the influence of such important aspects as the tactical design, the play structure, the opponent, the fitness of the players, timeouts, fouls, substitutions, etc. [10,12], so that it is of vital importance to record the highest number of variables possible to propose adapted and specific results for each individual. 

## 5. Conclusions

The present study has established the activity demands and speed profile in an U18 women’s basketball tournament. In spite of fatigue, it was found that on the third day players covered a greater distance at a higher intensity, mainly motivated by achieving a better final ranking. Regarding playing position, the forwards played for more minutes, covering a greater distance and performing more sprints and high intensity actions, thus providing the team with greater physical aptitude which can be determinant for the results of the match. 

### Limitations and Practical Applications

With this research the authors have tried to approach the activity demands and speed profile in u18 female players. The sample was collected in a single tournament, so the sample is small to generalize its results. Even so, with profiling, what is intended is to show how to create them with the aim of increase the individualizing of training for a specific player, team or competition. Profiling may be useful in player selection and development of sport-specific training programs [25]. A methodology is established for the approach of coaches to the creation of individual profiles and training plans and fine-tuning practice for their teams and players. 

## Figures and Tables

**Figure 1 ijerph-17-01477-f001:**
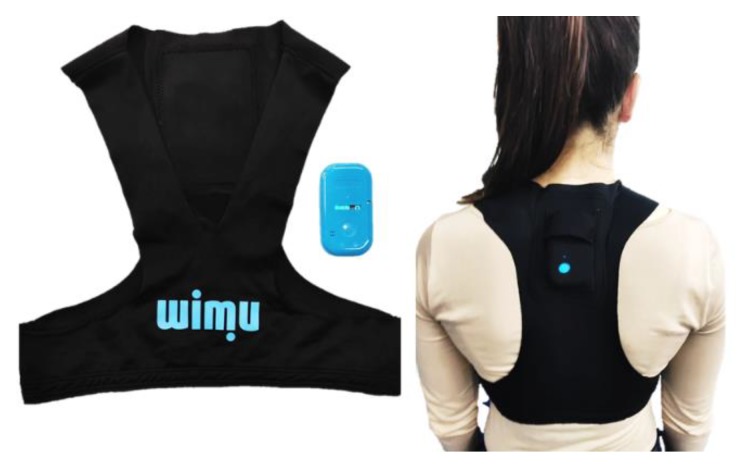
Player wearing the inertial device.

**Figure 2 ijerph-17-01477-f002:**
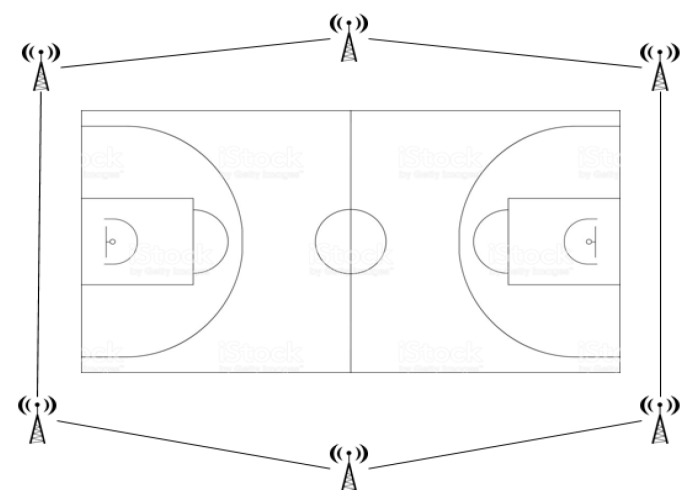
UWB system on basketball court.

**Table 1 ijerph-17-01477-t001:** Speed Ranges.

Intensity Levels	Speed
Standing	<3.6 km/h (<1 m·s^−1^)
Walking	3.6–6.5 km/h (1–1.81 m·s^−1^)
Jogging	6.5–10.2 km/h (1.81–2.83 m·s^−1^)
Running	10.2–14.4 km/h (2.83–4 m·s^−1^)
Sprinting	>14.4 km/h (>4 m·s^−1^)

**Table 2 ijerph-17-01477-t002:** Descriptive results by match day.

Variables	DAY 1		DAY 2		DAY 3		d
	Mean	Maximum	Mean	Maximum	Mean	Maximum	
Real time (minutes)	33.00	66.20	28.52	60.55	34.38	65.53	
Playing time (minutes)	18.25	37.02	18.69	38.10	19.57	35.22	
Distance (metres)	1995.27	4077.04	2118.36	4825.17	2388.61	4287.87	**b**
Distance/minute	112.67	310.73	113.74	170.73	123.96	304.25	**a, b**
Explosive Distance (metres)	235.71	547.83	270.12	586.04	292.72	534.79	
Explosive Distance/minute	14.00	42.66	15.14	34.56	15.48	34.04	**a, b**
Standing (metres)	389.90	893.35	355.85	823.89	417.03	885.92	
Standing (%)	20.15	37.87	17.22	32.74	17.13	24.30	
Walking (metres)	567.23	1228.98	562.72	1417.76	655.68	1266.07	
Walking (%)	27.52	36.00	26.65	50.61	26.79	35.02	
Jogging (metres)	439.14	904.53	495.34	1163.97	539.28	981.98	
Jogging (%)	21.51	27.46	22.86	28.59	22.73	29.13	
Running (metres)	403.22	909.28	487.77	1178.10	541.28	1115.27	**b**
Running (%)	20.18	35.52	22.25	30.41	22.72	31.81	
Sprinting (metres)	195.77	477.48	216.69	542.57	235.34	526.94	
Sprinting (%)	10.65	30.08	11.03	24.47	10.63	18.83	
N°. Sprints (n)	21.86	67.00	26.18	80.00	27.69	74.00	
Sprint/minute	1.56	11.10	1.49	3.70	1.51	2.98	
Sprint Distance (metres)	244.91	618.59	274.56	678.88	300.67	670.90	
Sprint Duration (seconds)	2.68	3.92	2.63	7.84	2.97	8.91	
Max Speed (km/h)	19.05	27.22	19.95	27.70	18.67	21.77	
Avg Speed (km/h)	5.05	6.78	5.42	6.81	5.41	6.28	**a, b**

*p* < 0.05. By Match Day: **a** (Match day 1 vs. Match day 2); **b** (Match day 1 vs. Match day 3); **c** (Match day 2 vs. Match day 3).

**Table 3 ijerph-17-01477-t003:** Descriptive results by playing position.

Variables	GUARD		FORWARD		CENTRE		d
	Mean	Maximum	Mean	Maximum	Mean	Maximum	
Real time (minutes)	27.00	58.53	41.07	66.20	31.14	65.53	**d**
Playing time (minutes)	16.11	35.22	23.56	38.10	18.67	32.72	**d, f**
Distance (metres)	1816.37	4077.04	2820.64	4825.17	2076.29	4347.47	**d, f**
Distance/minute	114.91	310.73	122.75	304.25	112.78	147.22	
Explosive Distance (metres)	233.16	471.99	351.61	586.04	231.20	509.36	**d, f**
Explosive Distance/minute	15.12	42.66	15.89	34.56	13.25	25.79	
Standing (metres)	326.05	830.03	497.09	893.35	380.10	885.92	**d, f**
Standing (%)	18.73	37.87	17.39	23.44	18.21	23.44	
Walking (metres)	495.42	1211.27	772.45	1417.76	579.68	1249.05	**d, f**
Walking (%)	26.81	50.61	26.96	32.59	27.38	35.02	
Jogging (metres)	425.04	1003.89	623.33	1163.97	462.84	1027.61	**d, f**
Jogging (%)	22.58	28.46	21.77	24.74	22.50	29.13	
Running (metres)	395.24	873.65	634.23	1115.27	448.34	1178.10	**d, f**
Running (%)	21.51	35.52	22.38	26.96	21.16	30.41	
Sprinting (metres)	174.61	477.48	293.53	542.57	205.33	457.08	**d, f**
Sprinting (%)	10.37	25.55	11.49	30.08	10.74	18.83	
N°. Sprints (n)	21.12	74.00	32.62	60.00	25.39	80.00	**d**
Sprint/minute	1.40	3.33	1.58	3.70	1.66	11.10	
Sprint Distance (metres)	217.76	618.59	375.35	678.88	261.45	579.31	**d, f**
Sprint Duration (seconds)	2.77	8.91	2.74	3.22	2.75	3.29	
Max Speed (km/h)	19.16	27.70	19.64	27.22	18.98	22.71	
Avg Speed (km/h)	5.25	6.27	5.39	6.81	5.25	6.28	

*p* < 0.05 By Playing Position: **d** (Guard vs. Forward); **e** (Guard vs. Centre); **f** (Forward vs. Centre).

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
