# Peer review of "Activity Demands and Speed Profile of Young Female Basketball Players Using Ultra-Wide Band Technology"

_ijerph, 2020, doi:10.3390/ijerph17051477_

Round 1

Reviewer 1 Report

I am full of admiration for the authors of the manuscript who, first of all, showed resourcefulness, using a relatively new measurement tool for their research, and secondly that they put a lot of work into performing the examination. I would not like to discourage them, although the methodological approach to the resulting manuscript is quite poor. First, the authors should verify the used research tool. Check its repeatability and validation. Repeatability and validation are the basis of scientific research and should be done first. If a test-retest has been carried out before the tests were carried out, then the authors should have given its results. If the research tool is widely used for scientific work and other researchers have already studied its repeatability, then authors should provide specific repeatability values ​​obtained by other authors. Otherwise, attempting to create a speed profile in women under 18 using a research tool with unknown repeatability, with an unknown minimal measurement error etc. and on such a small research group is absolutely impossible.

Line 2: female instead of women would sound better

Line 3: We write about specific measurement tool in the title only when it’s really crucial. The aim of Your study was to determine the speed profile in female basketball players under 18, so the ultra wide band technology is just a measurement tool. The name of the measurement tool would be useful if you want to provide information about speed profile in female basketball players under 18 determined by ultra wide band technology while the speed profile in female basketball players under 18 determined by other tool have been already existed. Or if your study was somehow concentrated on the measurement tool (reliability, validity, comparing to another device, creating new device etc.). However, you should place the name of the device in the abstract as for the people who would search for any kind of analysis done by ultra wide band.

The Abstract should be completely revised and rewritten. The conclusion should directly answer for the aim of the study. The aim of your study was to determine the speed profile of U18 female basketball players, and the main conclusion from the abstract is that depending on gender and age, competition in basketball causes different physiological adaptations. Also, don’t write time, distance and speed with a capital letter. Be more clear on describing the method. The results should include some important numbers.

The Introduction is far too long in comparison to a very short Discussion. You start with writing about the measurement tool, instead of introducing us into the performance profiles. Why had you decided to do your research? Was it because you had a clinical problem that was a lack of speed profiles in female basketball players U18 or was it because you had bought a new equipment and you want to write about it? Leave the comparison to other methods to the Discussion where you should compare your results with another authors.

Line 30: In which way they seemed to be contrasting?

Line 45-46: How do you know that this tool is objective and reliable? Did you or other authors check that?

Line 53-61 It’s a bit confusing. The first lines are about match activity generally in sports of any kind? Firstly you write about speed, then about percentage activity, and you conclude that the values differ. Well, they differ, because they are related with different features. Or do u mean that the range of percentage activity is so large because of lack of specificity and individualisation in different studies?

Line 68-69: This is the most important information in your introduction because it says why you want to do your research. And everything should be linked to this sentence and maybe even you should start with this sentence.

Line 78: The sentence about practical implications of your research should be in the end of the Discussion, not in the aim of the study.

Line 81: We always start with ethics and with ‘placing’ the research. So you should firstly write where and when the study was conducted. Then write about the local ethics committee approval, and what’s very important, I don’t think that in case of girls under 18, the players and couches consents are enough. As I know, in case of a person under 18, the consent of a parent/legal guardian should be gained.

When it comes to the studied material, I’m not really sure whether you can determine a speed profile in basketball players under 18 analysing only 13 guards, only 22 forwards etc. I don’t know whether it’s not too small group to make such a generalised study. If you have analysed more that one team, then maybe you could create a ‘profile’. But in this matter it’s more like a profile of this particular team. You should evaluate also other teams to see whether your profile is similar to their profile – and this is all about the reliability and validity.

I would add a photo of a player wearing the device and some actual photos of the device as not everybody may be familiar with how it looks.

Generally, the whole section concerning method and measurement tool should be revised according to the reliability of the measurement.

Line 237: the first sentence seems out of topic. It’s more like speaking in general, but it’s not really related with what you wrote in the abstract or as the aim of your study.

The Discussion lacks of the limitations of the study and practical implications.

Author Response

REVIEWER 1

Open Review

English language and style

( ) Extensive editing of English language and style required 
( ) Moderate English changes required 
( ) English language and style are fine/minor spell check required 
(x) I don't feel qualified to judge about the English language and style 

Yes

Can be improved

Must be improved

Not applicable

Does the introduction provide sufficient background and include all relevant references?

( )

( )

(x)

( )

Is the research design appropriate?

( )

( )

(x)

( )

Are the methods adequately described?

( )

( )

(x)

( )

Are the results clearly presented?

( )

( )

(x)

( )

Are the conclusions supported by the results?

( )

(x)

( )

( )

Comments and Suggestions for Authors

I am full of admiration for the authors of the manuscript who, first of all, showed resourcefulness, using a relatively new measurement tool for their research, and secondly that they put a lot of work into performing the examination. I would not like to discourage them, although the methodological approach to the resulting manuscript is quite poor. First, the authors should verify the used research tool. Check its repeatability and validation. Repeatability and validation are the basis of scientific research and should be done first. If a test-retest has been carried out before the tests were carried out, then the authors should have given its results. If the research tool is widely used for scientific work and other researchers have already studied its repeatability, then authors should provide specific repeatability values ​​obtained by other authors. Otherwise, attempting to create a speed profile in women under 18 using a research tool with unknown repeatability, with an unknown minimal measurement error etc. and on such a small research group is absolutely impossible.

Line 2: female instead of women would sound better

Done, we feel better using "female"

Line 3: We write about specific measurement tool in the title only when it’s really crucial. The aim of Your study was to determine the speed profile in female basketball players under 18, so the ultra wide band technology is just a measurement tool. The name of the measurement tool would be useful if you want to provide information about speed profile in female basketball players under 18 determined by ultra wide band technology while the speed profile in female basketball players under 18 determined by other tool have been already existed. Or if your study was somehow concentrated on the measurement tool (reliability, validity, comparing to another device, creating new device etc.). However, you should place the name of the device in the abstract as for the people who would search for any kind of analysis done by ultra wide band.

We have considered important to define the instrument used in the title. This is the first study that uses Ultra wided band technology to create performance profiles and one of the first to describe activity demands. With this, we try to give importance to the advancement of technology in sports sciences.  If you think it is really necessary that this does not appear in the title we will remove it, but please consider it. Under your recommendations, the device name has been assigned in the abstract.

The Abstract should be completely revised and rewritten. The conclusion should directly answer for the aim of the study. The aim of your study was to determine the speed profile of U18 female basketball players, and the main conclusion from the abstract is that depending on gender and age, competition in basketball causes different physiological adaptations. Also, don’t write time, distance and speed with a capital letter. Be more clear on describing the method. The results should include some important numbers.

We have tried to improve the abstract for a better understanding. You can see the changes marked in red in the text.

“Abstract: Performance profiles have begun to be identified in order to help coaches to individualize training according to the age and gender of the athlete. Therefore, the aim of this study was to determine the activity demands and speed profile of U18 female basketball players during competitive matches. Time variables (real and playing time), distance variables (distance performed, intensity zones, high intensity distance and distance covered sprinting) and speed variables (number of sprints, sprint duration, maximum speed and average speed) were recorded from forty-eight players belonging to four teams (13 guards, 22 forwards and 13 centers). WIMUPROTM inertial measurement units with ultra-wide band (UWB) indoor-tracking technology recorded six matches during final four in the season 2018/2019. A one factor ANOVA with Cohen’s effect sizes (d) were used to identify the differences between groups (playing position and match day). Distance per minute (123.96 vs 112.67 m), high intensity distance per minute (15.48 vs 14 m), running distance (403.2 vs 541.28 m) and average speed (5.05 vs 5.41 km/h) was significantly higher on day 3 than 1, respectively. About playing position, forwards played more minutes during games, so covered a greater distance, more sprints and high intensity actions than the rest. In spite of fatigue, day 3 showed a greater intensity than day 1, therefore, the last day was the crucial one for the teams in the tournament. Forwards when playing more minutes obtain higher absolute values ​​but not per minute which could mean a lower performance of the team”

The Introduction is far too long in comparison to a very short Discussion. You start with writing about the measurement tool, instead of introducing us into the performance profiles. Why had you decided to do your research? Was it because you had a clinical problem that was a lack of speed profiles in female basketball players U18 or was it because you had bought a new equipment and you want to write about it? Leave the comparison to other methods to the Discussion where you should compare your results with another authors.

We have followed your recommendations in the introduction. We have restructured the order and we have given more importance to profiling analysis. In addition, we have reduced the information on instruments used. We believe we have improved its quality and understanding for the reader.

We have added 2 new references about profiling:

Sampaio, J. (2013). Routledge handbook of sports performance analysis. T. McGarry, P. O'Donoghue, & A. J. de Eira Sampaio (Eds.). London: Routledge.

Ostojic, S. M., Mazic, S., & Dikic, N. (2006). Profiling in basketball: Physical and physiological characteristics of elite players. Journal of strength and Conditioning Research20(4), 740.

Line 30: In which way they seemed to be contrasting?

We have explain it better: “However, studies have reached conflicting conclusions due to the data processing [5]. The use of different analysis methodologies, fundamentally in the processing of playing time (total time vs. live time) has produced very disparate results, meaning that variables like distance are affected and hinder comparative analysis among studies. [6].”

Line 45-46: How do you know that this tool is objective and reliable? Did you or other authors check that?

We have added a reference about accuracy and inter-unit reliability about instrument used:

Bastida-Castillo, A., Gómez-Carmona, C. D., la Cruz-Sánchez, D., Reche-Royo, X., Ibáñez, S. J., & Pino Ortega, J. (2019). Accuracy and inter-unit reliability of ultra-wide-band tracking system in indoor exercise. Applied Sciences9(5), 939.

Line 53-61 It’s a bit confusing. The first lines are about match activity generally in sports of any kind? Firstly you write about speed, then about percentage activity, and you conclude that the values differ. Well, they differ, because they are related with different features. Or do u mean that the range of percentage activity is so large because of lack of specificity and individualisation in different studies?

We have tried to clarify the concepts developed in this paragraph:

“In previous investigations, match activity in basketball has been classified into the following different categories: Standing/walking (0-1.67 m·s-1); Jogging (1.1- 3.3 m·s-1); Running (3-7 m·s-1) and Sprinting (>7m·s-1) [4, 14-18]. According to the review by Stojanović [6] the percentages of metres performed in each velocity zone are established according to the following format: Standing/walking: 23.4-66.3%; Jogging: 5.6-36.3%; Running: 4.5-33.2%; and Sprinting: 0.3-8.5%. These range of values differ mainly due to the lack of specificity and individualization in the different studies. In women’s basketball, it has been reported that the players cover an average of 5214 ± 315 m per game. The intensity at which they work during the match is 30.2 -39.3% walking, 12.8-24% jogging, 4.9-11% running, and 0.6-7.8% sprinting [4, 15, 16, 18] being different from the results shown above in general basketball”

Line 68-69: This is the most important information in your introduction because it says why you want to do your research. And everything should be linked to this sentence and maybe even you should start with this sentence.

Dear reviewer, thank you for your suggestion, you are right. We have moved this at the beginning of the introduction and we believe that everything has been much better.

Line 78: The sentence about practical implications of your research should be in the end of the Discussion, not in the aim of the study.

We have deleted it and we have created a section of practical implications at the end of the document.

Line 81: We always start with ethics and with ‘placing’ the research. So you should firstly write where and when the study was conducted. Then write about the local ethics committee approval, and what’s very important, I don’t think that in case of girls under 18, the players and couches consents are enough. As I know, in case of a person under 18, the consent of a parent/legal guardian should be gained.

Dear reviewer, thank you for your suggestion, we have modified that; “All the players and coaches were informed about the research protocol, requisites, benefits, and risks, and the coach of each team written consent was obtained before the start of the study conforming to the Code of Ethics of the World Medical Association (Declaration of Helsinki) which was approved by the Ethics Committee of the University of Extremadura (nº 1234/2019)”

When it comes to the studied material, I’m not really sure whether you can determine a speed profile in basketball players under 18 analysing only 13 guards, only 22 forwards etc. I don’t know whether it’s not too small group to make such a generalised study. If you have analysed more that one team, then maybe you could create a ‘profile’. But in this matter it’s more like a profile of this particular team. You should evaluate also other teams to see whether your profile is similar to their profile – and this is all about the reliability and validity.

You are right. With this research the authors try to give information on how the players of this category behave in demands of activity and speed. Since nothing about it is found in the literature, we believe that this approach to characterization is very important. We have added in the limitations of the study that it is not a great sample, but the importance of stablish performance profiles is the individualization of players training plans and fine-tunning of practice.

We have added another 2 new references about profiling:

I would add a photo of a player wearing the device and some actual photos of the device as not everybody may be familiar with how it looks.

We have added a photo with player wearing the device:

Figure 1. Player wearing the inertial device

Generally, the whole section concerning method and measurement tool should be revised according to the reliability of the measurement.

We have tried to improve the method section for a better understanding of it and so that it can be replicated by the rest of the scientific community. You can see the changes marked in red in the text.

Line 237: the first sentence seems out of topic. It’s more like speaking in general, but it’s not really related with what you wrote in the abstract or as the aim of your study.

We have deleted it.

The Discussion lacks of the limitations of the study and practical implications.

A paragraph with limitations of the study and practical applications of the results has been added in the conclusions section with the aim that these results can be useful for coaches.

Limitations and practical applications

With this research the authors have tried to approach the activity demands and speed profile in u18 female players. The sample was collected in a single tournament, so the sample is small to generalize its results. Even so, with profiling, what is intended is to show how to create them with the aim of increase the individualizing of training for a specific player, team or competition. Profiling may be useful in player selection and development of sport-specific training programs. A methodology is established for the approach of coaches to the creation of individual profiles and training plans and fine-tuning practice for their teams and players. “

Reviewer 2 Report

It is a very interesting manuscript, due to the novelty in the individualized application of training based on the profiles of the players.
The theoretical framework clearly explains the state of the matter and shows the need for this type of work.
Problem that is approached with a methodology appropriate to the proposed objectives.
The results obtained are adequate are relevant, and the conclusions respond to the proposed objectives.
I think this manuscript should be published in this journal.

Author Response

REVIEWER 2

Open Review

English language and style

(x) Extensive editing of English language and style required 
( ) Moderate English changes required 
( ) English language and style are fine/minor spell check required 
( ) I don't feel qualified to judge about the English language and style 

Yes

Can be improved

Must be improved

Not applicable

Does the introduction provide sufficient background and include all relevant references?

(x)

( )

( )

( )

Is the research design appropriate?

(x)

( )

( )

( )

Are the methods adequately described?

(x)

( )

( )

( )

Are the results clearly presented?

(x)

( )

( )

( )

Are the conclusions supported by the results?

(x)

( )

( )

( )

Comments and Suggestions for Authors

It is a very interesting manuscript, due to the novelty in the individualized application of training based on the profiles of the players.
The theoretical framework clearly explains the state of the matter and shows the need for this type of work.
Problem that is approached with a methodology appropriate to the proposed objectives.
The results obtained are adequate are relevant, and the conclusions respond to the proposed objectives.
I think this manuscript should be published in this journal.

We have tried to improve the method section for a better understanding of it and so that it can be replicated by the rest of the scientific community. You can see the changes marked in red in the text.

Reviewer 3 Report

The submission was well written and presented in an easy to follow format. 

Please make these corrections to clear up a few points. 

Line 36. What does this statement refer to, “Following on from the above,…”. If referring to a study then state the study or is this paragraph a continuation of the previous one?

Lines 68-73. The authors are discussing stages of long-term athletic development from teen to adult age but should delineate whether this adult age is recreational, collegiate or elite (Professional or Olympic).

Lines 133-134. The term “competition” and “game” are used but previously “match” was term. Unless competition is referring to competition phase then just add phase.

Line 190. Was this study “the first” or was it just another review? The current sentence structure needs rewriting for clarity. “Ziv and Lidor [24] carried out a first literature review…”

Line 210. Add the word “accumulated” after distance or “total” or a term with similar context.

Author Response

REVIEWER 3

Open Review

English language and style

( ) Extensive editing of English language and style required 
( ) Moderate English changes required 
(x) English language and style are fine/minor spell check required 
( ) I don't feel qualified to judge about the English language and style 

Yes

Can be improved

Must be improved

Not applicable

Does the introduction provide sufficient background and include all relevant references?

(x)

( )

( )

( )

Is the research design appropriate?

(x)

( )

( )

( )

Are the methods adequately described?

(x)

( )

( )

( )

Are the results clearly presented?

(x)

( )

( )

( )

Are the conclusions supported by the results?

(x)

( )

( )

( )

Comments and Suggestions for Authors

The submission was well written and presented in an easy to follow format. 

Please make these corrections to clear up a few points. 

Line 36. What does this statement refer to, “Following on from the above,…”. If referring to a study then state the study or is this paragraph a continuation of the previous one?

References to these articles have been included:

Abdelkrim, N. B., El Fazaa, S., & El Ati, J. (2007). Time–motion analysis and physiological data of elite under-19-year-old basketball players during competition. British Journal of Sports Medicine, 41(2), 69-75.

Scanlan, A., Dascombe, B., & Reaburn, P. (2011). A comparison of the activity demands of elite and sub-elite Australian men's basketball competition. Journal of Sports Sciences, 29(11), 1153-1160

Lines 68-73. The authors are discussing stages of long-term athletic development from teen to adult age but should delineate whether this adult age is recreational, collegiate or elite (Professional or Olympic).

Improving the training processes during the formative stages has the objective that the players get to compete at a better and higher competitive level. Therefore, the phrase "and compete at high level in professional leagues" has been added at the end of the paragraph.

Lines 133-134. The term “competition” and “game” are used but previously “match” was term. Unless competition is referring to competition phase then just add phase.

Done. We have used “match”

Line 190. Was this study “the first” or was it just another review? The current sentence structure needs rewriting for clarity. “Ziv and Lidor [24] carried out a first literature review…”

We have deleted “first” and we have modified the sentence for clarity: “Ziv and Lidor [24] carried out a review of the literature in which they collected data on the demands of activity in women's basketball through time motion analysis”

Line 210. Add the word “accumulated” after distance or “total” or a term with similar context.

Done. We hace added “accumulated” after distance.

Reviewer 4 Report

The Authors describe a activity demands and speed profile of young women basketball.  I my opinion this research will be an important resource for other researchers in this area.

Suggested corrections:

In the abstract and in the purpose of the paper, the Authors write: „..the aim of this study was to determinate the speed profile…”, but the title of the study and the used variables suggest that not only speed parameters were analyzed. Please explain. In aim of the paper Authors write: „The findings from this research can be used to design conditioning programmes to optimise training.”, but I did not find practical or application conclusions. Please add them to paper. In chapter 2.2. please provide reference to Table 1. In chapter 2.3 the authors write that they use the mean and standard deviation. They do not use standard deviation in any table. Please correct. Please provide the limitations of the study. I propose to extend the discussion, this is the most important part of the work.

Author Response

REVIEWER 4

Open Review

English language and style

( ) Extensive editing of English language and style required 
( ) Moderate English changes required 
( ) English language and style are fine/minor spell check required 
(x) I don't feel qualified to judge about the English language and style 

Yes

Can be improved

Must be improved

Not applicable

Does the introduction provide sufficient background and include all relevant references?

( )

(x)

( )

( )

Is the research design appropriate?

(x)

( )

( )

( )

Are the methods adequately described?

( )

(x)

( )

( )

Are the results clearly presented?

( )

( )

(x)

( )

Are the conclusions supported by the results?

( )

( )

(x)

( )

Comments and Suggestions for Authors

The Authors describe a activity demands and speed profile of young women basketball.  I my opinion this research will be an important resource for other researchers in this area.

Suggested corrections:

In the abstract and in the purpose of the paper, the Authors write: „..the aim of this study was to determinate the speed profile…”, but the title of the study and the used variables suggest that not only speed parameters were analyzed. Please explain.

Dear Reviewer, we are very grateful for your suggestion since you are right. In the objective it is not clearly defined that the activity demands were also analyzed. We apologize for the mistake. We have modified the objective and added this: “the aim of this study was to determine the activity demands and speed profile of U18 female basketball players during competitive matches”

In aim of the paper Authors write: „The findings from this research can be used to design conditioning programmes to optimise training.”, but I did not find practical or application conclusions. Please add them to paper.

A paragraph on practical applications of the results has been added in the conclusions section

““Limitations and practical applications

With this research the authors have tried to approach the activity demands and speed profile in u18 female players. The sample was collected in a single tournament, so the sample is small to generalize its results. Even so, with profiling, what is intended is to show how to create them with the aim of increase the individualizing of training for a specific player, team or competition. Profiling may be useful in player selection and development of sport-specific training programs. A methodology is established for the approach of coaches to the creation of individual profiles and training plans and fine-tuning practice for their teams and players. “

In chapter 2.2. please provide reference to Table 1.

This reference is added in the Intensity Zones variable: "A K-means cluster analysis was used to distribute the total distance covered classified into the following speed zones of five intensity levels (Table 1): Standing, Walking, Jogging, Running and Sprinting, analyzing the meters covered in each group and the percentage based on the total meters. "

In chapter 2.3 the authors write that they use the mean and standard deviation. They do not use standard deviation in any table. Please correct.

Please provide the limitations of the study.

A paragraph on Limitations has been added in the conclusions section

Limitations and practical applications

With this research the authors have tried to approach the activity demands and speed profile in u18 female players. The sample was collected in a single tournament, so the sample is small to generalize its results. Even so, with profiling, what is intended is to show how to create them with the aim of increase the individualizing of training for a specific player, team or competition. Profiling may be useful in player selection and development of sport-specific training programs. A methodology is established for the approach of coaches to the creation of individual profiles and training plans and fine-tuning practice for their teams and players. “

I propose to extend the discussion, this is the most important part of the work. 

We have tried to improve the review, you can see the changes marked in red

Round 2

Reviewer 1 Report

The manuscript has been improved.